# HPV Vaccination as Adjuvant to Conization in Women with Cervical Intraepithelial Neoplasia: A Study under Real-Life Conditions

**DOI:** 10.3390/vaccines8020245

**Published:** 2020-05-23

**Authors:** Marta del Pino, Cristina Martí, Ines Torras, Carla Henere, Meritxell Munmany, Lorena Marimon, Adela Saco, Aureli Torné, Jaume Ordi

**Affiliations:** 1Institute Clinic of Gynecology, Obstetrics, and Neonatology, Hospital Clínic, 08036 Barcelona, Spain; marti@clinic.cat (C.M.); itorras@clinic.cat (I.T.); carlahenere@gmail.com (C.H.); mmunmany@clinic.cat (M.M.); atorne@clinic.cat (A.T.); 2Institut d’Investigacions Biomèdiques August Pi i Sunyer (IDIBAPS), Universitat de Barcelona, 08036 Barcelona, Spain; 3Department of Pathology, Hospital Clínic, University of Barcelona, 08036 Barcelona, Spain; lmarimon@clinic.cat (L.M.); masaco@clinic.cat (A.S.); jordi@clinic.cat (J.O.); 4Institute for Global Health (ISGlobal), Hospital Clínic, Universitat de Barcelona, 08036 Barcelona, Spain

**Keywords:** HPV vaccine, conization, HSIL/CIN2-3, persistent/recurrent disease

## Abstract

*Background:* Recent studies have shown preliminary evidence that vaccination against human papillomavirus (HPV) could decrease the risk of persistent/recurrent HSIL in women treated for high-grade cervical intraepithelial lesion (HSIL). We aimed to determine the benefits of HPV vaccination in patients undergoing conization for HSIL in real-life conditions and evaluate vaccination compliance associated with different funding policies. *Methods:* From January 2013 to July 2018, 265 women underwent conization in our center. From January 2013 to July 2017, treated patients (*n* = 131) had to pay for the vaccine, whereas after July 2017 the vaccine was publicly funded and free for treated women (*n* = 134). Post-conization follow-up controls were scheduled every six months with a Pap smear, HPV testing, and a colposcopy. *Results:* 153 (57.7%) women accepted vaccination (vaccinated group), and 112 (42.3%) refused the vaccine (non-vaccinated group). Persistent/recurrent HSIL was less frequent in vaccinated than in non-vaccinated women (3.3% vs. 10.7%, *p* = 0.015). HPV vaccination was associated with a reduced risk of persistent/recurrent HSIL (OR 0.2, 95%CI: 0.1–0.7, *p* = 0.010). Vaccination compliance increased when the vaccine was publicly funded (from 35.9% [47/131] to 79.1% [106/134], *p* < 0.001). *Conclusions:* HPV vaccination in women undergoing conization is associated with a 4.5-fold reduction in the risk of persistent/recurrent HSIL. Vaccination policies have an important impact on vaccination compliance.

## 1. Introduction

Women with high-grade squamous intraepithelial lesion/cervical intraepithelial neoplasia grade 2–3 (HSIL/CIN2-3) are treated by excisional procedure (conization) or ablation to prevent progression to cervical cancer [1]. However, persistent/recurrent disease occurs in 5–15% of the women treated for HSIL/CIN2-3 [2,3]. Moreover, these women are at higher risk of cervical cancer compared with the general population, even after adequate treatment [4,5,6,7]. Thus, long-term follow-up after treatment is recommended to detect persistent/recurrent lesions that might progress to cervical cancer [5,8,9,10].

Different mechanisms may contribute to this increased risk of cervical cancer. Persistent human papilloma virus (HPV) infection after treatment, with or without residual HSIL/CIN2-3, is clearly the most important established risk factor [11]. Another possible mechanism is the acquisition of a new HPV infection, since women who have already developed HSIL/CIN2-3 have shown an increased risk of developing new HPV-associated lesions [12]. This has been related to the continuation of lifestyle risk factors for HPV infection, which may remain throughout life, and to the higher susceptibility of these women to HPV persistence [7,13]. It is well established that women diagnosed and treated for pre-cancer are unable to control and clear HPV infection, and therefore, constitute a clear risk-group for persistent/recurrent disease [7,13].

In the last few years, five studies have reported preliminary evidence showing that HPV vaccination of women undergoing treatment for HSIL/CIN2-3 may result in a significant reduction of persistent/recurrent lesions during follow-up, [14,15,16,17,18] and that a high titer of antibodies against HPV reduces the risk of new infections and recurrent lesions. Moreover, some of these studies have also suggested a protective role of the HPV vaccine in women already harboring HPV infection [16], although the exact mechanism of this phenomenon and its clinical impact have not been fully elucidated.

Due to the results of the initial studies, our department started to actively recommend HPV vaccination to all women treated for HSIL in January 2013 [14,15,19]. From 2013 to June 2017, the cost of the vaccine had to be covered by the patients. After the publication of a consensus guideline supported by several national scientific societies recommending HPV vaccination in women treated for premalignant cervical lesions [19], in July 2017, the public health system of Catalonia started funding the HPV vaccine for women undergoing treatment [20]. This funding retrospectively covered all women who had undergone treatment for HSIL/CIN2-3 in the previous 12 months (between July 2016 and June 2017), and prospectively covered all patients undergoing conization. The cost of the vaccine was funded by the public health system in all these women.

The aim of this study was to compare outcomes after cervical conization in terms of persistent/recurrent HSIL in HPV-vaccinated and non-vaccinated women in a real-life situation, and evaluate vaccination compliance associated with the different funding policies.

## 2. Methods

### 2.1. Selection Criteria

All women undergoing conization between January 2013 and July 2018 were eligible for the study. Following the recommendations of the American Society for Colposcopy and Cervical Pathology and the Spanish Society of Cervical Pathology and Colposcopy [10,21,22,23], the criteria for conization were: (1) HSIL/CIN2-3 diagnosis in a colposcopy-directed biopsy and/or an endocervical curettage, and (2) repeated cytological result of HSIL in at least two Pap smears separated by six months in patients with a histological diagnosis of low-grade (L) SIL/CIN1 or no lesion, after excluding vaginal HSIL. All women had an HPV-positive testing prior to conization.

The exclusion criteria were: (1) diagnosis of cervical cancer in the conization specimen; (2) absence of follow-up after treatment, and (3) impossibility to record vaccination status.

During the study period, 310 women were treated by conization at our institution and were eligible for the study. In four patients the conization specimen showed cervical cancer, 12 were lost to follow-up, and in 29 the vaccination status was not recorded and they were, consequently, excluded. Thus, 265 women were finally included in the study.

### 2.2. Criteria for Treatment and Post-Treatment Follow-Up

Follow-up controls were scheduled every six months, up to at least 24 months. In each control, a Pap smear, HPV testing and a colposcopy, with colposcopy-directed biopsy or endocervical curettage, if indicated, were performed.

### 2.3. Liquid-Based Cytology and HPV Testing

Cervical samples were collected using a cytobrush and stored in PreservCyt solution (Hologic, Marlborough, MA, USA) for liquid-based cytology and HPV testing. A ThinPrep T2000 slide processor (Hologic) was used to prepare thin-layer cytology slides that were stained using the Papanicolaou method. The cytology slides were evaluated using the Bethesda nomenclature [24].

For HPV testing, the Cobas HPV test (Cobas 4800; Roche Molecular Diagnostics), based on a real-time polymerase chain reaction (PCR) system, was used. This method detects 14 high-risk HPV types and provides specific genotyping for HPV16 and HPV18.

### 2.4. Histological Diagnosis of Colposcopy-Directed Biopsy

Formalin-fixed, paraffin-embedded 4 μm sections were routinely stained with hematoxylin and eosin (H&E). All biopsies were stained for p16 using the CINtec histology kit (clone E6H4; mtm-Roche Laboratories, Heidelberg, Germany), following the manufacturer’s protocol. Immunohistochemical (IHC) staining was performed with the Autostainer Link 48 (Dako Co., Carpinteria, CA, USA), using the EnVision system (Dako). Cases with continuous block staining of cells of the basal and parabasal layers were considered as positive [25]. The histological diagnosis was established based on the combination of the H&E findings and the p16-stained sections. Biopsy specimens were classified as negative for SIL, LSIL/CIN1, HSIL/CIN2 or HSIL/CIN3 according to the Lower Anogenital Squamous Terminology (LAST) criteria [26].

### 2.5. Cervical Conization and Histological Diagnosis

After delineating the area of abnormality with acetic acid and iodine solution, 1 mL of 1% mupivacain was injected into each quadrant of the cervix. The loop size was selected according to the size of the area to be excised. Conization was performed under colposcopical vision, as described previously [27]. When endocervical involvement was suspected, a second selective endocervical sweep was performed using a smaller loop. On rare occasions in which an exocervical lesion was too large for excision in a single sweep, two or more systematic sweeps were performed. After the excision, selective coagulation of the bleeding areas was performed by ball diathermy.

Conization specimens were anatomically oriented, pinned to a cork support and fixed in 10% neutral buffered formalin. The excisional samples were thoroughly examined after processing the whole specimen in 3–14 paraffin blocks (median 6). Surgical margins were identified with ink and carefully examined. Both margin status (positive/negative) and margin location (exocervical/endocervical) were reported. Margins were considered negative if no SIL/CIN was detected. Positive margins were diagnosed when SIL/CIN of any grade was present.

### 2.6. HPV Vaccination Policy

From January 2013 to July 2017, (bivalent [2v] or quadrivalent [4v]) HPV vaccines were recommended by gynecologists to all women after the diagnosis of SIL/CIN requiring treatment. In this period, the vaccine was not funded by the public health system, and therefore, the patients who finally underwent vaccination had to pay for the vaccine. The 2v vaccine was scheduled at zero, one and six months, and the 4v vaccination was scheduled at zero, two and six months [28,29].

All women undergoing conization for HSIL/CIN2-3 after July 2017, when the Ministry of Health of Catalonia started funding HPV vaccination to treated women, were referred to the vaccination center of our institution for free administration of the nine-valent (9v) vaccine. In these women, the first dose of the vaccine was scheduled after HSIL/CIN2-3 diagnosis, and it was provided either immediately before or after conization, according to the availability of the vaccine in the vaccination department. In addition, following government legislation, all women who had undergone conization due to HSIL/CIN2-3 in the previous 12 months (from July 2016 to July 2017) and had not been previously vaccinated were called and free vaccination was offered. In these women, the first dose of the vaccine was administered 1–12 months after HSIL/CIN2-3 diagnosis, and the second and third doses were administered at months two and six, respectively [30].

Vaccination status of women was retrieved from clinical records.

### 2.7. Clinical Outcome Six Months after Conization

The status at the first post-conization control was categorized as follows: (1) persistent HSIL (histologically confirmed HSIL/CIN2-3); (2) persistent LSIL/HPV (abnormal cytology of any grade, and/or positive HPV test result, with biopsy diagnosis of LSIL/CIN1 or negative or no biopsy performed), and (3) no disease (negative HPV test, negative Pap test, and if available, negative biopsy).

### 2.8. Clinical Outcome at the End of the Follow-Up

The clinical outcomes of the patients at the end of the follow-up were categorized as follows: (1) persistent/recurrent HSIL (presence of histologically confirmed HSIL/CIN2-3, or a repeated HSIL result in at least two Pap smears separated by six months and positive HPV testing result, independently of the histological diagnosis); (2) persistent/recurrent LSIL/HPV (persistent abnormal cytological result of LSIL, ASC-US or AGUS, a single cytology result of HSIL and/or a positive HPV test result without histological diagnosis of HSIL/CIN2-3); and (3) no disease (negative HPV test, negative Pap test, and, if available, a negative biopsy).

The follow-up time was defined as the time from conization to the diagnosis of persistent/recurrent SIL or to the last recorded visit.

## 3. Data Analysis

The data were analyzed with the SPSS software (Version 25.0; SPSS, Inc, Chicago, IL, USA). Categorical variables are presented as absolute numbers and percentages and compared using the χ2 or Fisher exact test. Continuous variables are presented as mean and standard deviation (SD), and median and range. Means were compared using the analysis of variance test. Univariate logistic regression models were used to analyze the factors studied as independent risk factors for persistent/recurrent HSIL using the risk estimation as odds ratio (OR) with 95% confidence intervals (CI). In order to evaluate the influence among factors for a global model, we performed a multivariate approach calculating the adjusted OR (AOR) with the 95% CI by using factors with *p*-values ≤ 0.10 in the univariate models. *p* values < 0.05 were considered statistically significant.

## 4. Results

### 4.1. General Characteristics of the Women Included in the Study

The mean age of the 265 women included in the study was 39.8 years (SD 10.3). Of these, 131 were treated from January 2013 to June 2016, and 134 after July 2016 (76 from July 2016 to June 2017, and 58 from July 2017 to July 2018). Two hundred and forty-five patients (90.6%) were treated on the basis of histologically confirmed HSIL/CIN2-3 (149 [56.2%] with HSIL/CIN2, 91 [34.4%] with HSIL/CIN3), and 25 (9.4%) were treated based on a repeated cytological result of HSIL with negative or LSIL biopsy. The final histological diagnosis in the conization specimen was HSIL/CIN2-3 in 209/265 (78.9%) women (95 HSIL/CIN2, and 114 HSIL/CIN3), LSIL/CIN1 in 30/265(11.3%) and negative for SIL in 26/265 (9.8%). Among the 25 women treated due to repeated HSIL cytology with negative or LSIL biopsy, 9 (36.0%) showed HSIL/CIN 2-3 in the cone specimen (6 HSIL/CIN2 and 3 HSIL/CIN3), 10 had a histological diagnosis of LSIL/CIN1 in the cone specimen and in six no SIL/CIN lesion was found in the cone specimen. Ninety-seven women (36.6%) showed positive margins (44 involving only the exocervical margin, 45 involving only the endocervical margin, and eight involving both margins).

### 4.2. Characteristics of Vaccinated and Non-Vaccinated Women

Overall, 153 women (57.7%) accepted vaccination, and 112 (42.3%) refused the vaccine. Table 1 shows the clinical characteristics at the baseline visit, the histological diagnosis of the conization specimen and margin status for the vaccinated and non-vaccinated women. No differences were found between the two groups.

### 4.3. Vaccination Compliance and Vaccination Scheme

Vaccination compliance was 35.9% (47/131) for the women treated before June 2016, who had to pay for the vaccine themselves, and 79.1% (106/134) for the women treated after July 2016, who received free vaccination (*p* < 0.001). In this latter group, vaccine acceptance was 76.3% (58/76) for the women treated between July 2016 and June 2017, who were called 1–12 months after treatment and were invited to receive free vaccination, and 82.8% (48/58) for the women treated after July 2017, who were immediately referred to vaccination after the diagnosis of HSIL/CIN2-3 and the decision of treatment (*p* = 0.399).

Of the 153 women who were vaccinated, 10 (6.5%) received the first dose before the treatment and 143 (93.5%) after the treatment. All the women who were vaccinated before the treatment received the HPV vaccine between one and 12 months prior to conization. One hundred and eighteen of the 153 vaccinated women (77.1%) received the three doses, 16 (10.5%) received two doses, seven (4.6%) received only one dose and 12 women (7.8%) did not remember the number of doses received. No differences were identified in relation to the type of vaccine or the number of doses received in terms of clinical characteristics, the results of the baseline visit or histological diagnosis of the conization specimen (data not shown).

### 4.4. Results of the First Post-Conization Control

The mean time from treatment to the first post-conization control was 6.7 months (SD 4.4). The status at the first post-conization control was: persistent HSIL in 11 women (4.2%), persistent LSIL/HPV in 101 (38.1%; 35 had an abnormal cytology with a negative HPV test; 29 an abnormal cytology and a positive HPV test, and 37 positive HPV test result with normal cytology) and no disease in 153 (57.7%). No differences were found between vaccinated and non-vaccinated women in terms of persistent HSIL at the first post-conization control (8/153 [5.2%] vs. 3/112 [2.7%], respectively, *p* = 0.587).

Within the eight vaccinated women with persistent HSIL in the first post-conization control, one woman rejected a second treatment and showed persistent LSIL/HPV during the follow-up, one woman showed HSIL/CIN2-3 biopsy with a negative HPV test and underwent follow-up visits showing no disease at 18 months post-treatment. One 26-year-old woman with HSIL/CIN2-3 had a normal colposcopy. She was not treated and in the follow-up controls showed persistent HPV with no cytological or histological abnormalities, which persisted at the end of the follow-up (persistent/recurrent LSIL/HPV). Five women underwent a second conization procedure. Three of them showed no disease after the second treatment and two showed persistent/recurrent LSIL/HPV after the second procedure).

Within the non-vaccinated women, two rejected a second conization procedure. One of them had persistent/recurrent HSIL and the other persistent/recurrent LSIL/HPV at the end of follow-up. One woman underwent a second conization but developed a persistent/recurrent HSIL during follow-up.

### 4.5. Clinical Outcome at the End of Follow-Up

The mean follow-up was 22.4 months (SD 12.7; median 21.7 [range 8.0–77.2]). For the vaccinated women, the mean follow-up was 20.2 months (SD 11.9; median 18.5 [range 10.3–77.2]) and for the non-vaccinated women 25.3 months (SD 13.3; median 24.3 [range 8.0–59.9.2]] (*p* = 0.001). At the end of follow-up, 17/265 women (6.4%) had persistent/recurrent HSIL, 51 (19.3%) had persistent/recurrent LSIL/HPV infection (one had an abnormal cytology with a negative HPV test; 26 an abnormal cytology and a positive HPV test, and 24 positive HPV test result with normal cytology) and 197 women (74.3%) were considered as having no disease (negative HPV test, negative Pap test, and, if available, negative biopsy). There were no differences among the three outcome groups in terms of indication of conization, HPV genotype, cone diagnosis or status of margin status (data not shown). Figure 1 shows the women disposition of the women and the clinical outcomes at the end of follow-up according to vaccination status. The rate of persistent/recurrent HSIL was lower in vaccinated than non-vaccinated women (5/153 [3.3%] vs. 12/112 [10.7%], *p* = 0.015). The mean time from treatment to persistent/recurrent HSIL was 22.8 months (SD 13.5). Among the vaccinated women, no association was found between the type of vaccine or the number of doses and the prevalence of persistent/recurrent HSIL at the end of follow-up (*p* = 0.908 and *p* = 0.738, respectively). Table 2 shows the clinical and histological characteristics of the vaccinated and non-vaccinated women who developed persistent/recurrent HSIL at the end of follow-up.

Considering only the patients treated before July 2017, in whom a longer follow-up was available, persistent/recurrent HSIL was diagnosed in 4/105 (3.8%) vaccinated women and in 12/102 (11.8%) non-vaccinated women (*p* = 0.032). The mean follow-up for these women treated before July 2017 did not differ between vaccinated and non-vaccinated women (24.7 months [SD 11.7] vs. 26.8 months [SD 12.9], *p* = 0.210).

Table 3 shows the univariate and multivariate models for persistent/recurrent HSIL at the end of follow-up. In the univariate analysis, persistent LSIL/HPV and persistent HSIL at the first post-conization control significantly increased the risk of persistent/recurrent HSIL at the end of follow-up (OR: 4.1, 95%CI: 1.2–13.4; and OR: 14.9, 95%CI: 2.7–75.3, respectively, *p* = 0.001). HPV vaccination was also associated with a significantly reduced risk of persistent/recurrent HSIL at the end of follow-up (OR 0.3, 95%CI: 0.1–0.8, *p* = 0.021). Both factors remained significant in the multivariate analysis.

Table 4 shows the final outcome in vaccinated and non-vaccinated women according to the status at the first post-conization control. None of the 87 (0%) vaccinated women who showed no disease at the first post-conization control (negative HPV test, negative Pap test, and, if available, a negative biopsy) developed persistent/recurrent HSIL, whereas 4/65 (6.1%) of the non-vaccinated women had persistent/recurrent HSIL at the end of follow-up (*p* = 0.032). Persistent/recurrent HSIL was also lower in vaccinated women who showed persistent LSIL/HPV or persistent HSIL in the first post-conization control compared with non-vaccinated women, although the differences were not statistically significant.

## 5. Discussion

This is the first study to evaluate the effect of the HPV vaccine in women treated for HSIL in a real-life setting (i.e., with differences in dose compliance, time of administration and type of vaccine administered). The present study shows that the HPV vaccine is associated with a significant reduction of the risk of developing persistent/recurrent HSIL after conization. This effect was particularly clear in women who had no disease in the first post-conization control at six months (negative HPV test, negative Pap test, and, if available, a negative biopsy). Indeed, none of the vaccinated women who were free of SIL or HPV infection in the first post-conization control developed HSIL in the follow-up, thereby confirming that the vaccination had a clear effect on preventing the acquisition of a new HPV infection after treatment.

Four retrospective studies have suggested that the administration of the HPV vaccine to women who have undergone treatment for HSIL/CIN2-3 can reduce the risk of developing persistent/recurrent disease [8,9,10,11,12]. A recent study (SPERANZA) was the first to prospectively evaluate the effectiveness of HPV vaccination after conization in women treated for HSIL/CIN2-3. This study has shown that vaccination is associated with a 5-fold reduction in the risk of residual/recurrent disease [16]. In keeping with these previous findings, our study, which was prospective but conducted in a real-life clinical situation, showed a similar reduction (4.5-fold) in the risk of persistent/recurrent HSIL in vaccinated women.

The protective effect of the HPV vaccine was particularly clear in women who had no disease in the first post-conization control at six months (negative HPV test, negative Pap test, and, when performed, a negative biopsy): none of the vaccinated women who were free of disease in the first post-conization control developed HSIL in the follow-up, confirming that vaccination prevented the acquisition of new HPV infection after treatment. Interestingly, the women who had persistent LSIL/HPV infection or even HSIL in the first post-conization control also tended to have lower rates of persistent/recurrent HSIL at the end of follow-up when vaccinated, although the differences were not statistically significant (6.9% vs. 14.0%, *p* = 0.173; and 12.5% vs. 66.7%, *p* = 0.131 in vaccinated and non-vaccinated patients, respectively). No previous report has evaluated the risk of persistent/recurrent HSIL according to the status at the first post-conization control. Indeed, previous evidence indicates that HPV vaccines are eminently prophylactic, and have no clear efficacy against prevalent disease or HPV infection already present at the time of vaccination [31,32]. However, several studies have suggested that the HPV vaccine may have some benefits in women with prevalent HPV infection [16], as also suggested in our study, and have hypothesized that the HPV vaccine could evoke a local antibody effect that arrests the entrance of the virus into uninfected cells in the basal layer, preventing disease relapse.

Previous HPV vaccine studies in women treated of HSIL/CIN2-3 have used a single vaccine and have been designed under strict conditions in terms of vaccine doses [14,15,16,17]. In contrast, this series represents the real-life situation of HPV vaccination in an adult population, with different vaccines coexisting, and many women not completing the recommended three-dose immunization scheme. Although this study was not designed to evaluate the protective effect of the different types of HPV vaccine, it is interesting to note that the risk of persistent/recurrent HSIL was similar in all women regardless of whether they received the 2v, 4v or 9v vaccine. Finally, although HPV vaccines were initially studied in a three-dose schedule spaced at 0, 1–2 and 6 months, subsequent studies showed that a two-dose strategy spaced at 0 and 6–12 months had equivalent results in young adolescents [33], and recently, some studies have suggested an equivalent efficacy in preventing HSIL/CIN2-3 with a one-dose schedule [34,35,36]. All these studies were conducted in young pre-adolescent women and thus, these results cannot be extrapolated to a high-risk adult population. Interestingly, in our study, no differences were observed between women who received the complete three-dose immunization or only one or two doses of the HPV vaccine. These results indicate that two, or even one, HPV vaccine dose could be effective as well in adult women, and suggest that further investigation exploring the effect of vaccination schemes with a reduced number of doses might be of interest.

This study shows the effect of different funding policies on vaccination compliance. In the first period (from 2013 to June 2016 and some of the women treated from July 2016 to July 2017), the women had to cover the cost of the vaccines themselves, and although vaccination was recommended by the gynecologists, no recommendation was given in terms of type of vaccine. Consequently, the women were vaccinated with one of the two licensed vaccines (2v or 4v). In contrast, in the last period (from July 2016), all the patients received the 9v-vaccine funded by the public health system. The vaccination rates markedly increased from 36% in the initial period to 79% in the second period when the vaccine was publicly funded (*p* < 0.001). As there were no clinical or pathological differences between the non-vaccinated (most being treated in the first period) and the vaccinated women (most being treated in the second period), we can assume that the differences between vaccination compliance are due to the funding policies and not to any confounder factor. These previously poorly analyzed data are an interesting consideration for vaccine policies in high-risk groups.

In our study, the prevalence of positive margins in the conization specimen was 36.6%. Previous studies in non-vaccinated women have shown that women with positive margins in the conization specimen have a higher increased risk of developing persistent/recurrent disease than women with negative margins [37]. Interestingly, in the present series, all the vaccinated women, including those who had positive margins, showed a very low rate of persistent/recurrent HSIL, suggesting that the vaccine might have a protective role even in this high-risk subgroup of treated women.

The main strength of our study is that all the participants were intensively screened, with rigorous assessment of disease endpoints and procedures. However, there are some limitations that should be noted. One limitation is that women were vaccinated at different time points. Consequently, in each patient of this group the follow-up time included two different periods: one in which the woman had not been vaccinated yet, and therefore was not protected against HPV, and another in which the woman had been already vaccinated and was hypothetically protected. Due to the real-life nature of our study, the length of these two periods was highly variable from woman to woman. Thus, the cumulative risk of developing persistent/recurrent HSIL had to be evaluated by logistic regression instead of a Cox regression as the assumption of proportional hazards required in Cox models was not fulfilled. Another limitation is that the HPV testing method did not allow extensive HPV genotyping, thus, the effectiveness of the vaccine for the individual HPV genotypes could not be properly assessed. Finally, another possible limitation of the study is that biases in the vaccinated and non-vaccinated women due to differences in sexual history were not evaluated. Nevertheless, it is likely that the possible effects of these socio-behavioral parameters are limited, since no other clinical or pathological differences were found between the two groups. Finally, the mean length of follow-up was shorter in the group of vaccinated women compared with those who refused the vaccine. This is because the number of vaccinated women was much lower in the first period when the follow-up time was longer, and much higher in the second more recent period, when the follow-up was shorter (and the reverse for the non-vaccinated women). However, it is remarkable that when the analysis was restricted to the women treated before July 2017, for whom a long follow-up, similar for both groups was available, the vaccine also showed a reduction in the risk of persistent/recurrent HSIL (3.8% vs. 11.8%; *p* = 0.032).

## 6. Conclusions

In conclusion, the present series provides further evidence confirming the clinical benefits of HPV vaccination as an adjuvant to excision treatment for HSIL in real-life conditions, showing that HPV vaccination is associated with a 4.5-fold reduction in the risk of persistent/recurrent HSIL/CIN2-3 after conization. Additionally, our study emphasizes the importance of funding policies to achieve adequate vaccination compliance. Finally, vaccination strategies with less than the recommended three doses and the possible effect of the vaccine on patients with persistent disease or HPV infection after conization require further analysis in studies involving a larger number of patients.

## 7. Ethical Approval

The study was conducted according to the principles of the Declaration of Helsinki. Written informed consent was obtained from all patients for analysis of the samples (Biobank HPV collection R180220-194). For the present study, no additional procedures or test were required, and the management of the women included was performed following the clinical protocols of the hospital based on the current national guidelines. Thus, no formal IRB/ethics approval was required.

## Figures and Tables

**Figure 1 vaccines-08-00245-f001:**
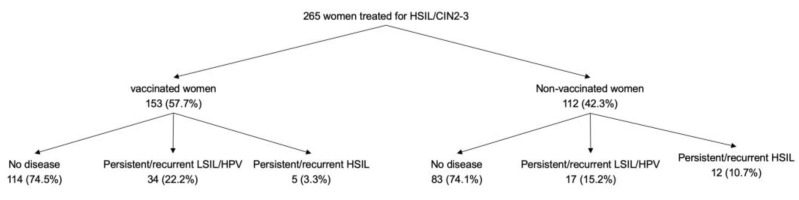
Disposition and clinical outcome at the end of follow-up of the women included in the study according to vaccination status.

**Table 1 vaccines-08-00245-t001:** Baseline clinical characteristics and results of the conization specimen in women who were vaccinated and those who refused the HPV vaccine. Values are absolute numbers and percentages.

Baseline Clinical Characteristics	Vaccinated Women (*n* = 153)		Non-Vaccinated Women (*n* = 112)		*p* *
Age			0.115
< 35 years	68	(44.4)	39	(34.8)	
≥ 35 years	85	(55.6)	73	(65.2)	
Indication of conization					0.300
Pap test HSIL & biopsy ≤ LSIL/CIN1	12	(7.8)	13	(11.6)	
Biopsy HSIL/CIN 2–3	141	(92.2)	99	(88.4)	
HPV genotype (pre-treatment)					0.063
HPV16 or 18	94	(61.4)	56	(50.0)	
Other HPV types (non 16 non 18)	59	(38.6)	56	(50.0)	
Year of treatment					<0.001
January 2013 to June 2016	47	(30.7)	84	(75.0)	
June 2016 to July 2018	106	(69.3)	28	(25.0)	
Conization specimen diagnosis					0.436
Negative	12	(7.8)	14	(12.5)	
LSIL/CIN1	17	(11.1)	13	(11.6)	
HSIL/CIN2-3	124	(81.1)	85	(75.9)	
Margin status					0.091
Negative	94	(61.4)	80	(71.4)	
Positive	59	(38.6)	32	(28.6)	
Type of HPV vaccine					
2v	30	(19.6%)	-	-	-
4v	7	(4.6%)	-	-	
9v	98	(64.1%)	-	-	
unknown	18	(11.8%)	-	-	

HPV: human papillomavirus. LSIL: low-grade squamous intraepithelial lesion; HSIL: high-grade squamous intraepithelial lesion: CIN: cervical intraepithelial neoplasia. 2v: bivalent. 4v: quadrivalent. 9v: ninevalent. * Fisher exact test.

**Table 2 vaccines-08-00245-t002:** Clinical characteristics, results at the baseline visit and of the first post-conization control performed at six months, and HPV genotype at the end of follow-up in the vaccinated and non-vaccinated women who showed persistent/recurrent HSIL at the end of follow-up.

Case Number	Age	Indication of Conization	Pre-Treatment HPV Genotype	Conization Diagnosis	Cone Margins	First Post-Conization Control Result (6 Months)	HPV Genotype at the End of Follow-Up	Vaccine Type	Doses
Vaccinated women who showed persistent/recurrent HSIL (*n* = 5)									
1	38.2	HSIL/CIN 2–3	HPVnon16/non18	LSIL/CIN1	Positive	Persistent LSIL/HPV	HPVnon16/non18	Unknown	3
2	41.2	Pap HSIL, biopsy LSIL/CIN1	HPVnon16/non18	HSIL/CIN2	Negative	Persistent LSIL/HPV	HPVnon16/non18	2v	3
3	41.6	HSIL/CIN 2–3	HPVnon16/non18	LSIL/CIN1	Positive	Persistent LSIL/HPV	HPVnon16/non18	9v	3
4	30.7	HSIL/CIN 2–3	HPV16	HSIL/CIN2	Positive	Persistent LSIL/HPV	HPV16	9v	3
5	46.5	Pap HSIL, biopsy LSIL/CIN1	HPV16	Negative	Negative	Persistent HSIL/CIN2-3	HPV16	9v	2
Non-vaccinated women who showed persistent/recurrent HSIL (*n* = 12)									
6	31.1	HSIL/CIN 2–3	HPVnon16/non18	HSIL/CIN2	Negative	No disease	HPV16	-	-
7	64.2	HSIL/CIN 2–3	HPVnon16/non18	HSIL/CIN2	Negative	Persistent LSIL/HPV	HPVnon16/non18	-	-
8	41.6	Pap HSIL, biopsy LSIL/CIN1	HPV16	LSIL/CIN1	Positive	Persistent LSIL/HPV	HPV16	-	-
9	45.4	HSIL/CIN 2–3	HPV16	HSIL/CIN3	Negative	No disease	HPVnon16/non18	-	-
10	33.0	HSIL/CIN 2–3	HPV18	HSIL/CIN2	Negative	No disease	HPVnon16/non18	-	-
11	66.2	HSIL/CIN 2–3	HPV16	HSIL/CIN3	Negative	Persistent LSIL/HPV	HPVnon16/non18	-	-
12	39.9	HSIL/CIN 2–3	HPVnon16/non18	HSIL/CIN3	Negative	No disease	HPVnon16/non18	-	-
13	33.2	HSIL/CIN 2–3	HPV16	HSIL/CIN2	Positive	Persistent LSIL/HPV	HPV16	-	-
14	50.4	HSIL/CIN 2–3	HPVnon16/non18	Negative	Negative	Persistent LSIL/HPV	HPVnon16/non18	-	-
15	61.4	Pap HSIL, biopsy LSIL/CIN1	HPVnon16/non18	LSIL/CIN1	Negative	Persistent HSIL/CIN2-3	HPVnon16/non18	-	-
16	54.9	HSIL/CIN 2–3	HPV16	LSIL/CIN1	Negative	Persistent HSIL/CIN2-3	HPV16	-	-
17	26.8	HSIL/CIN 2–3	HPV16	HSIL/CIN2	Positive	Persistent LSIL/HPV	HPV16	-	-

HPV: human papillomavirus. LSIL: low-grade squamous intraepithelial lesion; HSIL: high-grade squamous intraepithelial lesion; CIN1: cervical intraepithelial neoplasia grade 1; CIN2: CIN grade 2; 9v: nine-valent vaccine; 2v: bivalent vaccine.

**Table 3 vaccines-08-00245-t003:** Univariate and multivariate logistic regression analysis of risk factors associated with persistent/recurrent high-grade squamous intraepithelial lesion (HSIL). Persistent/recurrent HSIL was diagnosed based on the presence of histologically confirmed HSIL/CIN2-3, or a repeated HSIL result in at least two Pap smears separated by six months, independently of the histological diagnosis.

Variable	Persistent/Recurrent HSIL at the End of Follow-Up					
	Univariate Analysis			Multivariate Analysis		
	OR	(95% CI)	*p*	AOR	(95% CI)	*p*
Age			0.345			-
< 35 years	1			-	-	
≥ 35 years	1.67	(0.6–4.9)		-	-	
HPV type			0.753			-
HPV non 16 non 18	1			-	-	
HPV 16 and/or 18	0.85	(0.3–2.3)		-	-	
Margins of the conization specimen			0.932			-
Negative	1			-	-	
Positive	1.0	(0.4–2.9)		-	-	
First post-conization control status			0.001			0.018
No disease	1			1		
Persistent LSIL/HPV	4.1	(1.2–13.4)		4.3	(1.3–14.3)	
Persistent HSIL/CIN2-3	14.9	(2.7–75.3)		21.0	(3.6–123.5)	
HPV vaccination			0.021			0.010
No	1			1		
Yes	0.3	(0.1–0.8)		0.2	(0.1–0.7)	

HSIL/CIN2-3: high-grade squamous intraepithelial lesion/cervical intraepithelial neoplasia grade 2-3; OR: odds ratio; AOR: Adjusted OR; CI: confidence interval; HPV: human papillomavirus; LSIL: low-grade squamous intraepithelial lesion. Persistent/recurrent HSIL/CIN2-3 at the end of follow-up is defined as histologically confirmed HSIL/CIN2-3, or a repeated HSIL result in at least two Pap smears separated by six months and positive HPV testing result, independently of the histological diagnosis.

**Table 4 vaccines-08-00245-t004:** Persistent/recurrent disease at the end of follow-up in vaccinated and non-vaccinated women related to the first post-conization control status. Values are absolute numbers and percentages.

		Clinical Outcome at the End of Follow-Up		
Status at the First Post-Conization Control (6 Months)	No Disease	Persistent/Recurrent LSIL/HPV	Persistent/Recurrent HSIL	*p* *
No disease (*n* = 153)				0.032
Non-vaccinated	55 (83.3)	7 (10.7)	4 (6.1)	
Vaccinated	78 (89.7)	9 (10.3)	0 (0.0)	
Persistent LSIL/HPV (*n* = 101)				0.173
Non-vaccinated	28 (65.1)	9 (20.9)	6 (14.0)	
Vaccinated	33 (56.9)	21 (36.2)	4 (6.9)	
Persistent HSIL/CIN2-3 (*n* = 11)				0.131
Non-vaccinated	0 (0.0)	1 (33.3)	2 (66.7)	
Vaccinated	3 (37.5)	4 (50.0)	1 (12.5)	

LSIL: low-grade squamous intraepithelial lesion; HPV: human papillomavirus; HSIL/CIN2-3: high-grade squamous intraepithelial lesion/cervical intraepithelial neoplasia grade 2–3. Status at first control post-conization are defined as follows: (1) persistent HSIL (histologically confirmed HSIL/CIN2-3); (2) persistent LSIL/HPV (abnormal cytology of any grade, and/or positive HPV test result, with biopsy diagnosis of LSIL/CIN1 or negative or no biopsy performed), and (3) no disease (negative HPV test, negative Pap test, and if available, negative biopsy). (1) persistent/recurrent HSIL (presence of histologically confirmed HSIL/CIN2-3, or a repeated HSIL result in at least two Pap smears separated by six months and positive HPV testing result, independently of the histological diagnosis); (2) persistent/recurrent LSIL/HPV (persistent abnormal cytological result of LSIL, ASC-US or AGUS, a single cytology result of HSIL and/or a positive HPV test result without histological diagnosis of HSIL/CIN2-3); and (3) no disease (negative HPV test, negative Pap test, and, if available, a negative biopsy). * Fisher exact test.

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
