# Peer review of "HPV Vaccination as Adjuvant to Conization in Women with Cervical Intraepithelial Neoplasia: A Study under Real-Life Conditions"

_vaccines, 2020, doi:10.3390/vaccines8020245_

Round 1
Reviewer 1 Report
This manuscript described the effect of HPV vaccine in women treated for SIL/CIN in a real-life setting. It is interesting and will help the understanding of HPV vaccine for patients underwent conization with diagnosis of SIL. However, several points are incomplete or require improvement as shown below.
1. Although this study included the cases with no disease and LSIL/CIN1 by preoperative biopsy and conization, some cases were diagnosed as persistent/recurrent HSIL/CIN2-3. However, it is neither persistent nor recurrent. Furthermore, it is described that first sentence and the end of Discussion.
2. The sentence is incomplete at line 217.
3. Show the median and the range of follow-up time instead of the mean and SD for the better understanding at line 219.
3. In Table 3, it is described Cox model, but authors described logistic regression model in Materials and Methods. Which is correct?
4. This study did not perform HPV genotyping, but Cobas 4800 can identify the infection of HPV16 and 18. So, protective effect of HPV16/18 by HPV vaccine should be evaluated.
5. The sentence in page 10 at line 10 is incomplete.
Author Response
REVIEWER 1
QUESTION 1: Although this study included the cases with no disease and LSIL/CIN1 by preoperative biopsy and conization, some cases were diagnosed as persistent/recurrent HSIL/CIN2-3. However, it is neither persistent nor recurrent. Furthermore, it is described that first sentence and the end of Discussion.
RESPONSE:
As the reviewer points out, seven patients were considered as having persistent/recurrent HSIL/CIN2-3 at the end of follow-up although they did not show histological HSIL/CIN2-3 in the cone specimen: five had a LSIL/CIN1 and in two no lesion was identified in the cone. Thus, we understand that the use of the term “persistent/recurrent HSIL/CIN2-3” in these women could raise some concerns. However, as stated in the Methods section, the criteria for conization were: 1) HSIL/CIN2-3 diagnosis in a colposcopy-directed biopsy and/or an endocervical curettage, and 2) repeated cytological result of HSIL in at least two Pap smears separated by six months in patients with a histological diagnosis of low-grade (L) SIL/CIN1 or no lesion, after excluding vaginal HSIL. Consequently, in all these seven cases the conization was performed due to previous evidence of HSIL: four had a previous biopsy confirming HSIL/CIN2-3 and three had a persistent cytological result of HSIL, and vaginal lesions had been carefully excluded. Thus, we believe that the term “persistent/recurrent HSIL/CIN2-3” can be confidently used, as the existence of a preexisting HSIL lesion is well documented.
To clarify this issue, emphasizing that all women had a previous biopsy of HSIL/CIN2-3 or, at least, two Pap smears with HSIL result, we have modified the order of the inclusion criteria and moved the sentence on the criteria for conization to the first subsection of the Methods section (page 2, lines 73-78 of the revised section, underlining the sentences moved):
“All women undergoing conization between January 2013 and July 2018 were eligible for the study. Following the recommendations of the American Society for Colposcopy and Cervical Pathology and the Spanish Society of Cervical Pathology and Colposcopy [10,21–23], the criteria for conization were: 1) HSIL/CIN2-3 diagnosis in a colposcopy-directed biopsy and/or an endocervical curettage, and 2) repeated cytological result of HSIL in at least two Pap smears separated by six months in patients with a histological diagnosis of low-grade (L) SIL/CIN1 or no lesion, after excluding vaginal HSIL. The exclusion criteria were: 1) diagnosis of cervical cancer in the conization specimen; 2) absence of follow-up after treatment, and 3) impossibility to record vaccination status.”
In any case, we agree with the reviewer that some women underwent conization without histological confirmation of “HSIL/CIN2-3 prior to the conization. In our series 16/265 women (6.0%) underwent conization because of having a high probability of harboring an underdiagnosed HSIL/CIN2-3 lesion (women with persistent HSIL cytology and no vaginal lesions), but the conization specimen did not show HSIL/CIN2-3. Interestingly, the risk of persistent/recurrent HSIL does not differ if these women are excluded from the analysis. Following the reviewer’s recommendation, we have added a new sentence to the Results section to remark this point.
The revised version (page 4, lines 182-185, underlining the new sentence) now reads.
“The final histological diagnosis in the conization specimen was HSIL/CIN2-3 in 209/265 (78.9%) women (95 HSIL/CIN2, and 114 HSIL/CIN3), LSIL/CIN1 in 30/265(11.3%) and negative for SIL in 26/265 (9.8%). Among the 25 women treated due to repeated HSIL cytology with negative or LSIL biopsy, 9 (36.0%) showed HSIL/CIN 2-3 in the cone specimen (6 HSIL/CIN2 and 3 HSIL/CIN3), 10 had a histological diagnosis of LSIL/CIN1 in the cone specimen, and in 6 no SIL/CIN lesion was found in the cone specimen.”
Finally, following the reviewer’s suggestion, we have modified the first sentence and the end of the discussion to further clarify this issue.
The revised version (page 9, lines 320-321), crossing out the deleted text) now reads:
“This is the first study to evaluate the effect of the HPV vaccine in women treated for HSIL/CIN2-3 in a real-life setting”
The revised version (page 9, lines 405-406), crossing out the deleted text) now reads:
“In conclusion, the present series provides further evidence confirming the clinical benefits of HPV vaccination as an adjuvant to excision treatment for HSIL/CIN2-3 in real-life conditions”
QUESTION 2: The sentence is incomplete at line 217.
RESPONSE:
We apologize for the typo. We have corrected the mistake in the revised version (page 6, lines 241-242 in the revised version, crossing out the deleted words):
“One woman underwent a second conization but developed a persistent/recurrent HSIL during follow-up and”
QUESTION 3: Show the median and the range of follow-up time instead of the mean and SD for the better understanding at line 219.
RESPONSE:
Following the reviewer’s suggestion, the median and the range of follow-up have been added, in the Results section for better understanding (page 6, lines 244-246 in the revised version, underlining the new sentence and crossing out the deleted words):
“The mean follow-up was 22.4 months (SD 12.7; median 21.7 [range 8.0-77.2]). For the vaccinated women, the mean follow-up was 20.2 months for the vaccinated women (SD 11.9; median 18.5 [range 10.3-77.2]) and for the non-vaccinated women 25.3 months for the non-vaccinated women (SD 13.3; median 24.3 [range 8.0-59.9.2]] (p=0.001).”
This new presentation of the follow-up data has been added to the Methods section (page 4, lines 166-167 in the revised version, underlining the new words)
“Continuous variables are presented as mean and standard deviation (SD), and median and range. and Means were compared using the analysis of variance test.
QUESTION 4: In Table 3, it is described Cox model, but authors described logistic regression model in Materials and Methods. Which is correct?
RESPONSE:
We apologize for the typo. As described in the Material and Methods section, as well as in the legend of Table 3, the analysis showed in the table is the logistic regression analysis. The terms have been corrected in Table 3:
“Univariate Cox model” has been replaced by “Univariate analysis” and “Multivariate Cox model” has been replaced by “Multivariate analysis”
QUESTION 5: This study did not perform HPV genotyping, but Cobas 4800 can identify the infection of HPV16 and 18. So, protective effect of HPV16/18 by HPV vaccine should be evaluated.
RESPONSE:
We agree with the reviewer that data on the protective effect of the HPV vaccine on the types included in it is relevant. Effectively, Cobas 4800 provides data on HPV 16/18 genotyping. Indeed, the prevalence of HPV 16/18 pre-treatment was compared between vaccinated and non-vaccinated women, but the differences were not significant. Moreover, HPV16/18 was evaluated as a risk factor for persistent/recurrent HSIL in the univariate analysis, but these genotypes were not significantly associated with the risk of HSIL, and therefore this variable was not included in the multivariate analysis.
Despite recognizing the interest of this issue, the low number of HPV infections after treatment does not allow adequate evaluation of the analysis of this possible asociation. Thus, we did not delve further into this.
QUESTION 6: The sentence in page 10 at line 10 is incomplete.
RESPONSE:
We apologize for the typing error. Following the reviewer’s advice, we have completed the sentence in the revised version (page 10, lines 359-360, underlining the new sentence)
“Although this study was not designed to evaluate the protective effect of the different types of HPV vaccine, it is interesting to note that the risk of persistent/recurrent HSIL was similar in all women….”
Reviewer 2 Report
Dear Authors
I have read the manuscript entitle “HPV vaccination as adjuvant to conization in women with cervical intraepithelial neoplasia: a real-life study”. This study shows 2 main results: the first is that HPV vaccination is associated with a reduced risk of persistent/recurrent HIS, and the second is that vaccination policies have an important impact on vaccination compliance.
I have some doubts.
Title
After reading your work, I think that the kind of study was a Cohort Study or Prospective Observational Study. In any case, it was under real-life conditions. Please amend the title.
General Questions
The vaccine is available since 2008, but, you do not mention the vaccination status of these women before conization. You only mention that their status was retrieved from clinical records but only the vaccinated after conization. What do you think about women vaccinated prior to conization? Could this fact affect your findings?
Data Analysis section
The results of the influence among factors for a global model through a multivariate approach are shown by an adjusted OR (AOR). Please mention this AOR in this section and in table 3.
According to Data Analysis, continuous variables were compared using the analysis of variance test but I think this variance test was not applied.
Result section
In line 164 I suggest writing numbers instead of words for numbers. Unify criteria.
General characteristics of women should be included in Table 1.
Tables
Table 1: Mention the performed test for obtaining the p-value (Fisher exact-test).
Table 2: Could be column 1 “patients” instead of “patient”? Your sample size is 265 but table 2 only show 17. So I understand that the size of each group should appear in this first column called “patient”. I suggest calling this column “frequency”.
Table 3: I miss the frequency of each group.
A Cox model is a survival model but at the data analysis section you mention a logistic regression. The outcomes of these methods, Cox and Logistic regression are shown by OR, but I feel that the frequency of the data collection and the absence of variable time that a logistic regression was performed. Please explain.
Table 4: Mention the performed test for obtaining the p-value (Fisher exact-test).
Kind regards
Author Response
REVIEWER 2
QUESTION 1: Title
After reading your work, I think that the kind of study was a Cohort Study or Prospective Observational Study. In any case, it was under real-life conditions. Please amend the title.
RESPONSE:
Following the reviewer’s suggestion, we have amended the title. The new title now reads as follows (underlining the new words and crossing out the deleted words):
“HPV vaccination as adjuvant to conization in women with cervical intraepithelial neoplasia: a study under real-life conditions study”
QUESTION 2: General Questions
The vaccine is available since 2008, but, you do not mention the vaccination status of these women before conization. You only mention that their status was retrieved from clinical records but only the vaccinated after conization. What do you think about women vaccinated prior to conization? Could this fact affect your findings?
RESPONSE:
We agree with the reviewer that this is an important consideration. As described in the Results section, 10 out of the 153 vaccinated women (6.5%) had received the vaccine before undergoing treatment (page 5, lines 209-210 in the revised version). All these women received the vaccine between one and 12 months before conization and, therefore, they were included in the “vaccinated” group”. The low number of women vaccinated before conization (and especially before HSIL diagnosis) precludes any analysis on the effect of time of vaccine administration and the risk of persistent/recurrent HSIL
The information on the women vaccinated before conization has been added to the Results section (page 4, lines 210-211 in the revised version, underlining the newly added text):
“Of the 153 women who were vaccinated, 10 (6.5%) received the first dose before the treatment and 143 (93.5%) after the treatment. All the women who were vaccinated before the treatment received the HPV vaccine between one and 12 months prior to conization”
QUESTION 3: Data Analysis section
The results of the influence among factors for a global model through a multivariate approach are shown by an adjusted OR (AOR). Please mention this AOR in this section and in table 3.
RESPONSE:
Following the reviewer’s recommendation, we have mentioned the adjusted OR (AOR) in the Data Analysis section (page 4, lines 170-142 in the revised version, underlining the new words):
“In order to evaluate the influence among factors for a global model, we performed a multivariate approach calculating the adjusted OR (AOR) with the 95% CI using factors with p-values ≤ 0.10 in the univariate models. p values< 0 .05 were considered statistically significant.”
In table 3, the term AOR has also been used for the multivariate analysis.
QUESTION 4: Data Analysis section
According to Data Analysis, continuous variables were compared using the analysis of variance test but I think this variance test was not applied.
RESPONSE:
As described in the Methods section, the analysis of variance test was used to compare continuous variables such as mean follow-up data (page 6, lines 244-246 in the revised version):
“The mean follow-up was 22.4 months (SD 12.7; median 21.7 [range 8.0-77.2]). For the vaccinated women, the mean follow-up was 20.2 months (SD 11.9; median 18.5 [range 10.3-77.2]) and for the non-vaccinated women 25.3 months (SD 13.3; median 24.3 [range 8.0-59.9.2]] (p=0.001).”
And also, in page 8, lines 276-278 in the revised version:
“The mean follow-up of the women treated before July 2017 did not differ between vaccinated and non-vaccinated women (24.7 months [SD 11.7] vs. 26.8 months [SD 12.9], p=0.210).”
QUESTION 5: Result section
In line 164 I suggest writing numbers instead of words for numbers. Unify criteria.
RESPONSE:
Following the reviewer’s recommendation, words have been replaced by numbers (page 4, lines 175-179 in the revised version, underlined the numbers replacing the text):
The mean age of the 265 women included in the study was 39.8 years (SD 10.3). Of these, 131 were treated from January 2013 to June 2016, and 134 after July 2016 (76 from July 2016 to June 2017, and 58 from July 2017 to July 2018). 245 patients (90.6%) were treated on the basis of histologically confirmed HSIL/CIN2-3 (149 [56.2%] with HSIL/CIN2, 91 [34.4%] with HSIL/CIN3), and 25 (9.4%) were treated based on a repeated cytological result of HSIL with negative or LSIL biopsy.
QUESTION 6: Result section
General characteristics of women should be included in Table 1.
RESPONSE:
Following the reviewer’s suggestion, in table 1 we have included the only general characteristic that was not included in the original version: the year of treatment (before or after July 2016, when the vaccine was funded). Following the reviewer’s recommendation, the number and percentage of vaccinated and non-vaccinated women treated before and after July 2016 have been added to the table. Among the vaccinated women, 69.3% (106/153) were treated after July 2016 vs. 25.0% (28/112) of the non-vaccinated women (p< 0.001)
QUESTION 7: Tables
Table 1: Mention the performed test for obtaining the p-value (Fisher exact-test).
RESPONSE:
Following the reviewer’s suggestion, the test used for obtaining p-values (Fisher exact-test) has been mentioned in table 1
QUESTION 8: Tables
Table 2: Could be column 1 “patients” instead of “patient”? Your sample size is 265 but table 2 only show 17. So, I understand that the size of each group should appear in this first column called “patient”. I suggest calling this column “frequency”.
RESPONSE:
Table 2 describes the details of each particular patient who had persistent/recurrent HSIL at the end of follow-up (i.e. five among the vaccinated women and 12 among the non-vaccinated women). The first column of the table only indicates the case number of each patient.
In order to improve the clarity of table 2, we have replaced “Patient” by “Case number” as the heading for the first column.
In addition, to avoid misunderstanding with the sample size in table 2, we have modified the text of the headings (underlining the newly added text):
“Vaccinated women who showed persistent/recurrent HSIL (n=5)”
“Non-vaccinated women who showed persistent/recurrent HSIL (n=12)”
QUESTION 9: Tables
Table 3: I miss the frequency of each group.
RESPONSE:
Table 3 shows the univariate and multivariate logistic regression analyses for the factors possibly associated with persistent/recurrent HSIL. The frequency of these factors in the two groups of women (vaccinated and non-vaccinated women) is shown in Table 1. We do not show the global frequency of these factors in table 3 in order not to repeat this information.
QUESTION 10: Tables
A Cox model is a survival model but at the data analysis section you mention a logistic regression. The outcomes of these methods, Cox and Logistic regression are shown by OR, but I feel that the frequency of the data collection and the absence of variable time that a logistic regression was performed. Please explain.
RESPONSE:
As commented in QUESTION 4 - REVIEWER 1, this was a typo. The analysis shown in table 3 is a logistic regression analysis, as described in the Methods section, and the legend of the table. The term “Cox model” has been deleted in Table 3: “Univariate Cox model” has been replaced by “Univariate analysis” and “Multivariate Cox model” has been replaced by “Multivariate analysis”.
QUESTION 11: Tables
Table 4: Mention the performed test for obtaining the p-value (Fisher exact-test).
RESPONSE:
Following the reviewer’s suggestion, the test used for obtaining p-values (Fisher exact-test) has been mentioned in table 4.
Reviewer 3 Report
The manuscript is well written and very useful for the clinical setting.
- It would be better to explain the participant disposition as a Figure.
- How many participants underwent 2v or 4v vaccination in the study? Please add the information in the Table1. Also, is there any difference between the two groups for the outcome of the study?
- The margin positive late in participants seems very high. Please define what positive means. If the conization specimen diagnosis is CIN2, the margin is also CIN2. Or CIN1?
- As the authors stated in the discussion, the margin positive participants may not receive the merit of vaccination after the conization. Therefore, it would be better to discuss this outcome further in the text.
Author Response
REVIEWER 3
QUESTION 1: It would be better to explain the participant disposition as a Figure.
RESPONSE:
Following the reviewer’s recommendation, a new figure (Figure 1) has been added to the manuscript to show the disposition of the patients included in the study.
“Figure 1. Disposition and clinical outcome at the end of follow-up of the women included in the study according to their vaccination status.”
In addition, a new sentence has been added to the “Clinical outcome at the end of follow-up” subsection of the Results section (page 6, lines 252-254 in the revised version, underlining the new sentence).
“There were no differences among the three outcome groups in terms of indication of conization, HPV genotype, cone diagnosis or margin status (data not shown). Figure 1 shows the disposition of the women and the clinical outcomes at the end of follow-up according to vaccination status. The rate of persistent/recurrent HSIL was lower in vaccinated than non-vaccinated women (5/153 [3.3%] vs. 12/112 [10.7%], p=0.015).”
QUESTION 2: How many participants underwent 2v or 4v vaccination in the study? Please add the information in the Table1. Also, is there any difference between the two groups for the outcome of the study?
RESPONSE:
This information was already included in the initial version of the manuscript (Results section, previous version, page 4, lines 143-145):
“Among the vaccinated women, 30 (19.6%) received the 2v vaccine, 7 (4.6%) the 4v vaccine and 98 (64.1%) the 9v vaccine. Eighteen women (11.8%) did not remember which vaccine they had received.”
However, following the reviewer’s recommendation, this information has been deleted from the text and has been added to Table 1.
There were no differences between the type of the vaccine and the risk of prevalent/recurrent HSIL as stated in the “Clinical outcome at the end of follow-up” subsection of the Results section. This information was already included in the original version of the manuscript (page 6, lines 256-258 in the revised version):
“Among the vaccinated women, no association was found between the type of vaccine or the number of doses and the prevalence of persistent/recurrent HSIL at the end of follow-up (p=0.908 and p=0.738, respectively)”
QUESTION 3: The margin positive late in participants seems very high. Please define what positive means. If the conization specimen diagnosis is CIN2, the margin is also CIN2. Or CIN1?
RESPONSE:
The margins were considered positive if CIN was detected, independently of the grade of CIN. Only conization specimens with no CIN in the margins were considered as having “negative margins”. A new sentence has been added to the “Cervical conization and histological diagnosis “subsection of the Methods section to clarify this issue (page 3, lines 128-130 in the revised version, underlining the new words).
“Both margin status (positive/negative) and margin location (exocervical/endocervical) were reported. Margins were considered negative if no SIL/CIN was detected. Positive margins were diagnosed when SIL/CIN of any grade was present.”
QUESTION 4: As the authors stated in the discussion, the margin positive participants may not receive the merit of vaccination after the conization. Therefore, it would be better to discuss this outcome further in the text.
RESPONSE:
In our study, the prevalence of positive margins in the conization specimen was 36.6%. Previous studies in non-vaccinated women have shown that women with positive margins in the conization specimen have a higher risk of developing persistent/recurrent disease than women with negative margins. Interestingly, in our study, even vaccinated women who had positive cone margins had a very low rate of persistent/recurrent HSIL, suggesting that the vaccine might have a protective role even in this high-risk subgroup of treated women. However, the relatively low number of cases does not allow the detection of statistically significant differences on the possible impact of the margins of the conization between vaccinated and non-vaccinated women. Following the reviewer’s recommendation, we have added this information to the Discussion section (page 10, lines 383-389 in the revised version, underlining the new sentence).
In our study, the prevalence of positive margins in the conization specimen was 36.6%. Previous studies in non-vaccinated women have shown that women with positive margins in the conization specimen have a higher risk of developing persistent/recurrent disease than women with negative margins [37]. Interestingly, in the present series, all the vaccinated women, including those who had positive margins, showed a very low rate of persistent/recurrent HSIL, suggesting that the vaccine might have a protective role even in this high-risk subgroup of treated women.
Finally, a new reference has also been added to the Discussion section to support this statement (reference 37 in the revised version).
[37] L. Chen, L. Liu, X. Tao, L. Guo, H. Zhang, L. Sui, Risk Factor Analysis of Persistent High-Grade Squamous Intraepithelial Lesion after Loop Electrosurgical Excision Procedure Conization, J. Low. Genit. Tract Dis. 23 (2019) 24–27. https://doi.org/10.1097/LGT.0000000000000444.
Round 2
Reviewer 1 Report
The authors revised the paper according to the reviewer's comment. However, there is one point to be improved as shown below.
In Table 3, Cox proportional hazards regression analysis is adequate for univariate and multivarite analyses because time to persistent/recurrent HSIL and follow-up period were variable.
Author Response
QUESTION 1: In Table 3, Cox proportional hazards regression analysis is adequate for univariate and multivariate analyses because time to persistent/recurrent HSIL and follow-up period were variable.
RESPONSE:
In our study the follow-up time included, in each patient of the vaccinated group, two different periods: one in which the woman had not been vaccinated yet, and therefore was not protected against HPV, and another in which the woman had been already vaccinated and was hypothetically protected. Due to the real-life nature of our study, the length of these two periods was highly variable from woman to woman. Although it could be analyzed as a time-dependent covariable, we think it is a serious drawback for the assumption of proportional hazards. For this reason, we considered that the cumulative risk of developing persistent/recurrent HSIL would be a valid alternative to avoid the possible biases due to the abovementioned different hazards during the follow-up period when there are no right-censored cases (all patients completed the follow-up).
A new sentence has been added in the “Discussion” section to clarify this issue (lines 396-403, Page 11, underlined the newly added text)
“The main strength of our study is that all the participants were intensively screened, with rigorous assessment of disease endpoints and procedures. However, there are some limitations that should be noted. One limitation is that women were vaccinated at different time points. Consequently, in each patient of this group the follow-up time included two different periods: one in which the woman had not been vaccinated yet, and therefore was not protected against HPV, and another in which the woman had been already vaccinated and was hypothetically protected. Due to the real-life nature of our study, the length of these two periods was highly variable from woman to woman. Thus, the cumulative risk of developing persistent/recurrent HSIL had to be evaluated by logistic regression instead of a Cox regression as the assumption of proportional hazards required in Cox models was not fulfilled. Another limitation is that the HPV testing method did not allow extensive HPV genotyping, thus, the effectiveness of the vaccine for the individual HPV genotypes could not be properly assessed. Finally, another possible limitation of the study is that biases in the vaccinated and non-vaccinated women due to differences in sexual history were not evaluated.”
Reviewer 2 Report
Figure 1 should improve its readability.
Author Response
QUESTION 1: Figure 1 should improve its readability.
RESPONSE:
Following the reviewer suggestion, we have improved the readability of the Figure 1. A new Figure 1 has been added to the manuscript